# ReCoS: A Novel Benchmark for Cross-Modal Image-Text Retrieval in Complex Real-Life Scenarios

## ABSTRACT

Image-text retrieval stands as a pivotal task within information retrieval, gaining increasing importance with the rapid advancements in Visual-Language Pretraining models. However, current benchmarks for evaluating these models face limitations, exemplified by instances such as BLIP2 achieving near-perfect performance on existing benchmarks. In response, this paper advocates for a more robust evaluation benchmark for image-text retrieval, one that embraces several essential characteristics. Firstly, a comprehensive benchmark should cover a diverse range of tasks in both perception and cognition-based retrieval. Recognizing this need, we introduce ReCoS, a novel benchmark specifically designed for cross-modal image-text retrieval in complex real-life scenarios. Unlike existing benchmarks, ReCoS encompasses 12 retrieval tasks, with a particular focus on three cognition-based tasks, providing a more holistic assessment of model capabilities. To ensure the novelty of the benchmark, we emphasize the use of original data sources, steering clear of reliance on existing publicly available datasets to minimize the risk of data leakage. Additionally, to strike a balance between the complexity of the real world and benchmark usability, ReCoS includes text descriptions that are neither overly detailed, making retrieval overly simplistic, nor under-detailed to the point where retrieval becomes impossible. Our evaluation results shed light on the challenges faced by existing methods, especially in cognition-based retrieval tasks within ReCoS. This underscores the necessity for innovative approaches in addressing the complexities of image-text retrieval in real-world scenarios.

## CCS CONCEPTS

• **Information systems** → **Evaluation of retrieval results**; **Multimedia and multimodal retrieval**.

## KEYWORDS

Image-text Retrieval, Cross-modal Retrieval, Evaluation Benchmark

**ACM Reference Format:**
Anonymous Author(s). 2023. ReCoS: A Novel Benchmark for Cross-Modal Image-Text Retrieval in Complex Real-Life Scenarios. In *Proceedings of the 32nd ACM International Conference on Multimedia (MM'24), October 28-November 1, 2024, Melbourne, Australia.* ACM, New York, NY, USA, 10 pages. https://doi.org/XXXXXXX.XXXXXXX

**Unpublished working draft. Not for distribution.**

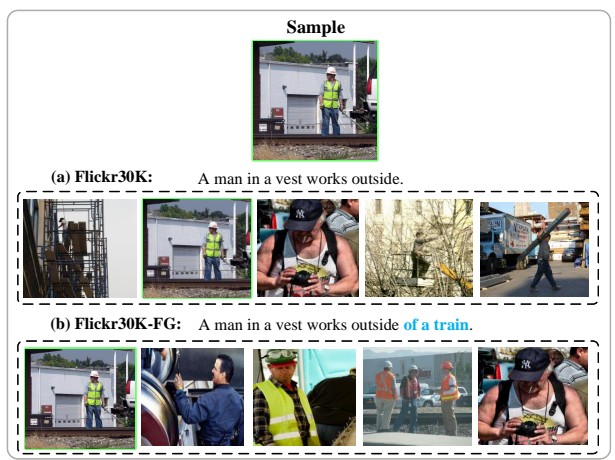

**Figure 1: Comparison of the five most similar images in both Flickr30K and Flickr30K-FG to a sample in the original Flickr30K dataset.**

## 1 INTRODUCTION

Image-text retrieval, as a fundamental and crucial problem in information retrieval, has attracted extensive attention in recent years. It aims to bridge the heterogeneous modality gap and achieve semantical matching by bidirectional retrieval, usually consisting of two subtasks: image-to-text (i2t) retrieval and text-to-image (t2i) retrieval. Text-to-image retrieval aims to search the target image from the whole candidate image pool given the text query, while image-to-text retrieval requires the model to search at least one target text description from the candidate text pool given the image query [3].

Early works such as VSE++ [9] and DPC [43], mapping images and text via convolutional neural networks, only roughly capture modalities' global correspondence, lacking fine-grained vision-language interaction. Consequently, researchers begin to focus on fine-grained retrieval [12, 26, 27, 42], such as SCAN [14], which uses Faster R-CNN [30] to encode images more finely and aligns image objects with sentence words. Recently, Visual Language Pretraining models such as BLIP-2 [18] and X2-VLM [40], acquiring alignment knowledge from image-text pairs through self-supervised tasks, have been used for image-text retrieval tasks and achieved near-perfect performance on widely used image-text retrieval benchmarks, i.e., MSCOCO-Test-5K and Flickr30K-Test-1K.

The benchmarks MSCOCO [22] and Flickr30k [37] have established themselves as widely recognized standards for image-text retrieval assessments. Chen et al. [3] brought attention to challenges associated with "coarse-grained" images and texts, including issues like small retrieval pool sizes and insufficient text descriptions, limiting the evaluation of fine-grained cross-modal semantic matching.

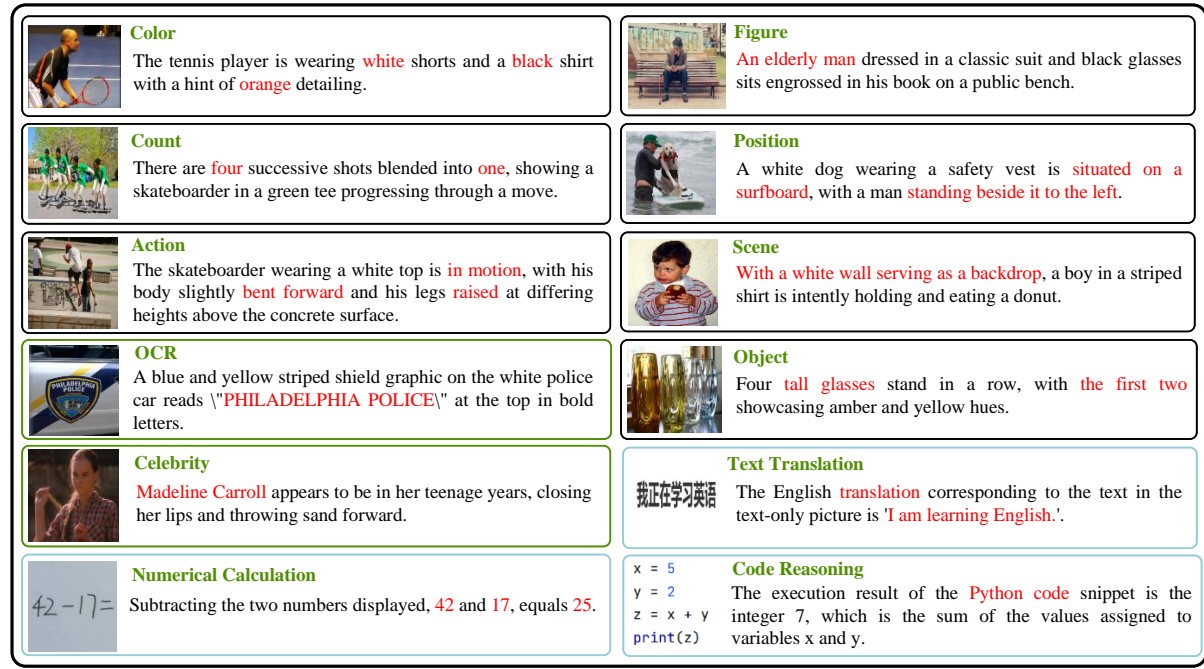

**Figure 2: Illustrative examples representing 12 sub-tasks within our $ReCoS_{v1}$ benchmark.**

In response, refined benchmarks, MSCOCO-FG and Flickr30k-FG, were introduced, aimed at improving the initial coarse-grained images and text descriptions. However, our manual evaluation revealed that MSCOCO-FG and Flickr30k-FG have reduced task difficulty due to the introduction of overly detailed text descriptions. For instance, as illustrated in Figure 1, the inclusion of the phrase "of a train" significantly simplifies the retrieval of the sample.

Recognizing the rapid advancements in Visual-Language Pre-training models, existing image-text retrieval benchmarks encounter challenges in effectively assessing the capabilities of these models. Notably, instances such as BLIP2 achieving near-perfect performance on MSCOCO and Flickr30K underscore the limitations of current benchmarks. To address these concerns and stay abreast of the dynamic landscape of image-text retrieval tasks, the imperative development of a new, comprehensive image-retrieval evaluation benchmark becomes apparent.

We posit that a comprehensive evaluation benchmark for image-text retrieval tasks should exhibit the following four characteristics:

(1) **Comprehensive Coverage**: Encompassing a wide array of subtasks, going beyond the recognition of specific objects. This includes aspects such as existence, count, position, color, and the composition of perceptual information with knowledge in Large Language Models (LLM) to handle more complex retrieval tasks, such as OCR retrieval, code image retrieval, landmark building retrieval, and celebrity retrieval.

(2) **Original Data Sources**: Avoiding reliance on existing publicly available datasets to minimize the risk of data leakage and ensure the novelty of the benchmark.

(3) **Balanced Text Descriptions**: Striking a balance in text descriptions, avoiding being under-detailed to the point

where retrieval becomes impossible and steering clear of over-detailed, making retrieval overly simplistic.

(4) **Alignment with Real-World Complexity**: Reflecting the intricacies of real-world scenarios to accurately gauge the capabilities of Visual-Language Pretraining models in diverse and challenging environments.

In pursuit of these objectives, we have curated a comprehensive evaluation benchmark for cross-modal image-text retrieval in complex real-life scenarios, aptly named ReCoS. The main contributions of this paper include:

(1) We review current image-text retrieval benchmarks and delineate four essential characteristics that a comprehensive evaluation benchmark for image-text retrieval tasks should possess.

(2) As illustrated in Figure 2, ReCoS comprises 12 sub-tasks in the domain of image-text retrieval. These tasks encompass coarse-grained recognition-based retrieval, covering seven types of retrieval tasks that emphasize aspects such as color, count, location, figure, object, action, and scene. Additionally, ReCoS incorporates fine-grained recognition-based retrieval tasks, including Optical Character Recognition (OCR) and the identification of celebrities. The benchmark extends to cognition-based retrieval, involving common numerical calculations, code reasoning-based retrieval to assess the model's ability to recognize and execute simple code, and text translation-based retrieval to evaluate proficiency in understanding multiple languages.

(3) We have created three benchmark versions: $ReCoS_{v0}$, $ReCoS_{v1}$, and $ReCoS_{v2}$. The construction of $ReCoS_{v1}$ and $ReCoS_{v2}$ are

built upon their predecessors, introducing heightened task difficulties.

(4) We evaluated several representative image-text retrieval models on our new benchmarks and further analyzed their capabilities in complex real-life scenarios.

In conclusion, our datasets serve as a significant catalyst for research, prompting the exploration of innovative approaches that seamlessly integrate both perceptual and cognitive aspects. The goal is to achieve more comprehensive and nuanced retrieval results, driving advancements in the field of multimodal retrieval systems.

## 2 RELATED WORK

### 2.1 Image-Text Retrieval Methods

In recent years, image-text retrieval has been extensively studied, and existing works can be roughly categorized into two categories. **Non-pretraining models.** Early works [7, 33, 34, 36, 44], such as VSE++ [9] and DPC [43], mapping images and text via convolutional neural networks, only roughly capture modalities' global correspondence, lacking fine-grained vision-language interaction. Consequently, researchers begin to focus on fine-grained retrieval [12, 26, 27, 42], such as SCAN [14], which uses Faster R-CNN [30] to encode images more finely and aligns image objects with sentence words. Subsequent non-pretraining models basically continue the ideas of VSE++ and SCAN, mainly improving the image-text alignment methods. The main directions of improvement include: 1) Optimizing image encoding methods. For example, the VSRN model [20] encodes image information more finely and uses Graph Convolutional Networks (GCN) [23] and Gated Recurrent Units (GRU) [1] to perform local-global semantic reasoning. 2) Improving the attention mechanism for image-text alignment. BFAN [23] proposes a bidirectional focal attention mechanism to eliminate the impact of irrelevant parts in images and text on the calculation of similarity scores. GSMN [24] uses a more complex way to align image-text, first constructs the relationship graphs of images and text separately, and then matches at the node level and structure level based on the two relationship graphs. SGRAF[6] utilizes GCN to capture the alignment between local and global information, and proposes an attention mechanism based on similarity filtering (SAF). 3) Introducing external knowledge. KASCE [31] and SGM [35] construct external scene graphs to enhance visual relationship learning. 4) Improving retrieval efficiency. Pan et al. [28] believe that the redundant part of cross-modal attention alignment is meaningless, and propose a fine-grained cross-modal alignment network (FCA-Net) to improve the efficiency of image-text retrieval. LexLIP [25] explores a Lexicon-Based cross-modal retrieval method, effectively reducing the computational cost of retrieval. **Pretraining models.** Visual Language Pretraining is designed to acquire visual language alignment knowledge from a vast collection of image-text pairs through self-supervised tasks. The structure of these models can be broadly categorized into single-stream, dual-stream, and hybrid models.

Single-stream models, such as UNITER [4] and OSCAR [21] [10] [41], employ a single encoder to process both images and text, facilitating the learning of alignment between visual and language information. UNITER utilizes a multi-layer Transformer encoder as its core component. OSCAR, in addition to basic image-text pairs, incorporates object tags on images to serve as a connection between the two modalities.

Dual-stream models, like CLIP [29] [8] [13], leverage two independent encoders for processing images and text. The features are then fused and aligned in subsequent stages. CLIP's image encoder may be based on ResNet or ViT structure, while the text encoder follows the Transformer structure. CLIP achieves large-scale contrastive learning pretraining on 400M image-text pairs, demonstrating zero-shot effectiveness on various tasks.

To leverage the strengths of both structures, hybrid models have emerged [2]. Models such as ALBEF [19] and BLIP [18] excel in modality alignment learning. ALBEF incorporates a multimodal encoder at the back, while BLIP, an enhanced version of ALBEF, retains ITC and ITM pretraining tasks but replaces the MLM task with the LM task to generate image descriptions. BLIP integrates text encoders and decoders for these tasks, sharing parameters across corresponding structural layers. The upgraded version, BLIP-2 [17], establishes a connection between the visual and text large models through a lightweight Q-Former model. In the domain of image-text retrieval, BLIP-2 has achieved nearly perfect state-of-the-art results.

### 2.2 Image-Text Retrieval Datasets

MSCOCO [22] and Flickr30k [37] are prominent datasets in image-text retrieval. Flickr30k comprises 31,783 images depicting everyday activities, each with five crowd-sourced descriptions. This dataset has 29,783 training images, 1,000 validation images, and 1,000 test images. On the other hand, MSCOCO is larger, with 123,287 images covering 91 common object categories. Each MSCOCO image has manually added bounding boxes, segmentation, and five descriptions. The dataset includes 113,287 training images, 5,000 validation images, and 5,000 test images.

However, Chen et al. [3] highlighted challenges associated with "coarse-grained" images and texts. They identified two primary issues: 1) small image retrieval pool sizes, resulting in significant variation or semantic sparsity, making retrieval targets easily distinguishable without the need for fine-grained semantic understanding; 2) a considerable number of text descriptions lacking detail. These challenges impose limitations on evaluating a model's fine-grained cross-modal semantic matching capabilities. In response, Chen et al. introduced two benchmarks, MSCOCO-FG and Flickr30k-FG, by refining the initially coarse-grained images and text descriptions in the original MSCOCO and Flickr30k datasets.

Nevertheless, the existing datasets mentioned above fall short in capturing intricate real-life scenarios, including code recognition and numerical computation. This inadequacy poses a challenge for conducting a thorough evaluation of model performance. In this paper, we introduce a pioneering benchmark tailored for image-text retrieval within complex real-life contexts. Our benchmark addresses this limitation by encompassing a wider array of sub-tasks within intricate scenarios, such as OCR retrieval, code image retrieval, landmark building retrieval, and celebrity retrieval. This diverse set of challenges enables a more exhaustive evaluation of model capabilities. The proposed benchmark stands as a rigorous assessment tool for image-text retrieval, particularly well-suited for evaluating the effectiveness of Visual Language Pretraining models.

# 3 BENCHMARK

In this section, we present the process of building our *ReCoS* benchmark, comprising three versions: $ReCoS_{v0}$, $ReCoS_{v1}$, and $ReCoS_{v2}$. We initiated by selecting datasets from diverse domains outside of image-text retrieval, including synthetically generated datasets, to mitigate data leakage risks. Subsequently, we executed four key steps in a scholarly manner, as illustrated in Figure 4. The following Section 3.1 first involves curating candidate images from these datasets to meet the specific requirements of the 12 subtasks. Section 3.2 delineates the detailed annotation process applied to the selected images. Section 3.3 discusses the introduction of additional similar images and the generation of perplexing textual descriptions to enhance the difficulty of the image-text retrieval task. These descriptions closely mirror those of the original images (see the appendix for details). Section 3.4 elaborates on the further augmentation of task complexity by introducing additional similar images devoid of annotations. Section 3.5 delineates the disparities in image quantity and subtask difficulty among the three versions.

## 3.1 Image Collection

We initially collected 500 images to create the candidate image set $\mathcal{D}$, as depicted in Figure 3, to address the following 12 types of retrieval tasks:

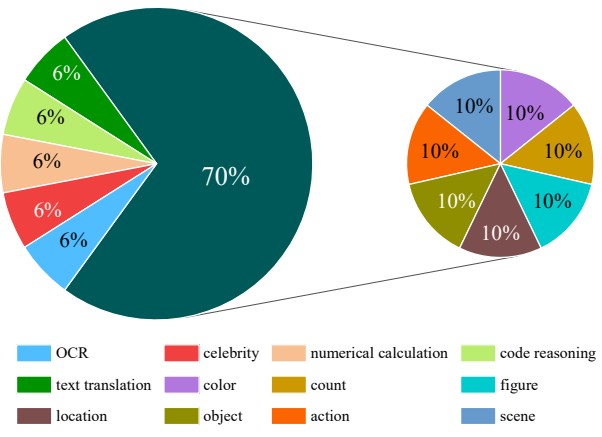

**Figure 3: Distribution of samples across 12 retrieval tasks within the candidate image set $\mathcal{D}$.**

(1) **Coarse-Grained Recognition-Based Retrieval.** This category involves seven types of retrieval tasks focusing on the recognition of color, count, location, figure, object, action, and scene. Each task type includes 50 images, totaling 350 images. Specifically, we select images related to these seven categories and generate corresponding descriptions based on the specific categories, ensuring that the descriptions covered different categories of words. The images for these tasks were gathered from COCO [1] with only images.

(2) **Fine-Grained Recognition-Based Retrieval.** This category includes two types of retrieval tasks focusing on Optical

[1] https://cocodataset.org/

Character Recognition (OCR) with images sourced from Totoltext [5] and the recognition of celebrities with images manually captured from videos on public websites. OCR is also a task for testing the basic capabilities of the model. The focus is on text recognition in pictures. Due to the diversity of scenes, OCR is still very difficult [38]. Tasks that include celebrities examine whether the model can specifically identify celebrities in the picture. Each task contains 30 images, totaling 60 images.

(3) **Numerical Calculation-Based Retrieval.** This category comprises 30 manually created images, predominantly featuring common numerical calculation scenarios, including simple addition, subtraction, multiplication, division calculations, and area calculations of two-dimensional images. Huang et al. [11] proposed that if large models are to move towards general artificial intelligence, they must have strong multi-modal cognitive capabilities. The primary objective of this category is to assess the model's ability to recognize handwritten digits and its reasoning proficiency for simple numerical calculation problems.

(4) **Code Reasoning-Based Retrieval.** In this category, there are 30 images of code generated by GPT-4, manually modified and verified. At present, there are some researches on code generation [32], but there is still a lack of research on the understanding of image code recognition. The primary task is to assess the model's capability to recognize code images and its understanding of simple code.

(5) **Text Translation-Based Retrieval.** There are already many large models that support multiple languages, and we hope to verify their translation capabilities. This category is established to assess the model's understanding of both Chinese and English languages. All 30 images in this category are manually designed. Given the current capabilities of existing models for Chinese and English, our current version focuses on designing simple Chinese-English translation tasks.

## 3.2 Image Annotation

**Generation of Image Descriptions.** Refinement of the process outlined in the dashed box in Figure 4 encompasses the following four steps applied to the candidate image set $\mathcal{D}$ to form a simple version image-text pair dataset $ReCoS_{v0}$:

- **Generate image description.**
  (1) For a given sample image $x \in \mathcal{D}$, generate five text descriptions $\{t_1(x), \cdots, t_5(x)\}$ using either GPT-4 or manual annotation.
- **Retrieve Top-k images.**
  (1) For each $t \in \{t_1(x), \cdots, t_5(x)\}$, employ the BLIP2 model to compute embeddings for both $t$ and the images in $\mathcal{D} - x$.
  (2) Identify the top $k$ most similar images $\mathcal{S}(t)$ from $\mathcal{D} - x$ based on the highest cosine similarities with $t$.
- **Validate the discriminability of text description.**
  (1) Compute $dis(t)$ to evaluate the discriminability of the text description $t$.
  (2) Select text descriptions that necessitate regeneration based on discriminability evaluation.
- **Output the refined description.**

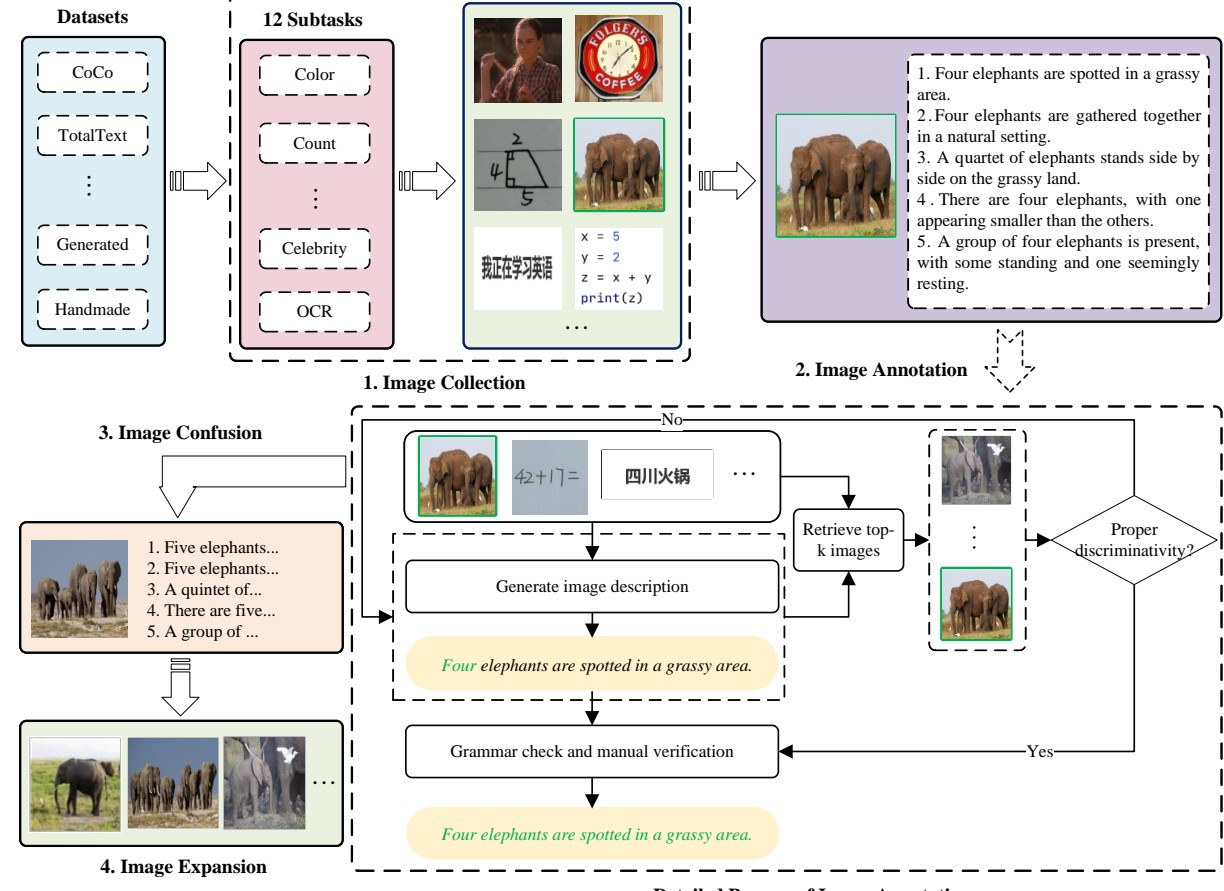

**Figure 4: Building Process of ReCoS Benchmark.**

(1) Conduct a grammar check and manual verification to ensure the quality of the text description.

(2) Output the 5 refined descriptions for each image.

**Validation of of text description.** In the aforementioned process, evaluating the discriminativity of the text description is a crucial step. This assessment is achieved by computing the discriminativity of the text description $t$ as $dis(t) = \text{entropy}(\text{softmax}(\text{sim}(t, \mathcal{S}(t))))$, where:

(1) $\text{sim}(t, \mathcal{S}(t))$ calculates the cosine similarities among $t$ and the images in $\mathcal{S}(t)$.

(2) softmax(.) generates classification probabilities using the softmax function with temperature parameters set to 1.

(3) entropy(.) outputs the entropy of the probability distribution.

The value of $dis(t)$, ranging between 0 and $\log k$, serves as an indicator of BLIP2's confidence in the retrieval result. The minimum value of 0 signifies high certainty in the retrieval, suggesting an overly simplistic retrieval task associated with an over-detailed text description. On the other hand, the maximum value of $\log k$ indicates uncertainty in the retrieval, suggesting an overly difficult retrieval task linked to an under-detailed text description.

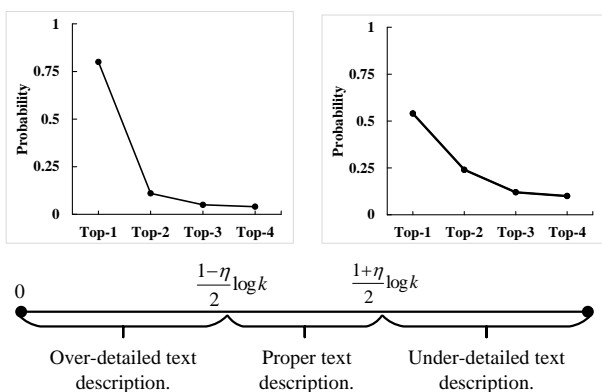

**Figure 5: Text description validation using discriminativity $dis(t)$ and selection of text descriptions necessitating regeneration via parameter $\eta$.**

The introduction of a parameter, denoted as $\eta \in (0, 1)$, plays a crucial role in the selection of text descriptions for regeneation.

Specifically, we identify descriptions where $dis(t) \notin [\frac{1-\eta}{2} \log k,$ $\frac{1+\eta}{2} \log k]$. text descriptions falling outside this range are considered unqualified, indicating the need for regeneration. By adjusting the value of $\eta$, we provide a mechanism to either expand or narrow the scope for identifying unqualified text descriptions. This flexibility empowers the annotator to fine-tune the discriminativity threshold according to the desired level of certainty in the retrieval task. A larger $\eta$ not only widens the range for identifying unqualified descriptions but also provides nuanced control over the delicate balance between task simplicity and difficulty in the annotation process. This parameter serves as a versatile tool, enabling customization based on the specific requirements and intricacies of the annotation task.

## 3.3 Image Confusion

In pursuit of heightened task difficulty, we augmented the challenge by introducing similar images paired with corresponding text descriptions. Specifically, we appended an additional 500 image-text pairs to $ReCoS_{v0}$, thereby creating a more intricate dataset denoted as $ReCoS_{v1}$. Our approach involves a semi-automated method, characterized by the following three main steps:

- **Similar image selection.** For each image $x \in \mathcal{D}$, we identified a similar image $x'$ from the source $x$ was sampled from. BLIP2 was employed to compute the image embedding, facilitating the selection of the most similar image.
- **Text description generation for similar images.** Utilizing GPT-4, we generated proper text descriptions $t'$ for the selected similar images $x'$. This process included the following steps:
  (1) Generate descriptions by referring to $t$ (original description).
  (2) Generate descriptions focusing on color, quantity, and other attributes.
  (3) Attempt to replace core attributes/keywords in provided descriptions based on image details.
  (4) Minimize word changes from the referenced description. Ensure each generated description is grammatical, logical, and uniquely detailed.
- Conduct manual checks and verifications that the generated similar description matches the similar image.

## 3.4 Image Expansion

To enhance the complexity of our benchmark, we implemented an image expansion strategy that involved introducing more intricate images into the dataset. This was achieved by incorporating additional images that exhibit similarities to the original ones but lack corresponding text descriptions. For each image $x \in \mathcal{D}$, we identified twenty similar images from the same source as $x$, resulting in a set of 20000 images. These images were then merged with $ReCoS_{v1}$, and the resulting set underwent refinement to eliminate duplicates, yielding $ReCoS_{v2}$ with a total of 15982 images. It is important to note that out of these, only 1000 images are paired with corresponding text descriptions.

## 3.5 Data Statistics

**Table 1: Three different versions of our ReCoS benchmark**

| Dataset | #Images | #Captions | #Categories | Difficulty level |
|---------|---------|-----------|-------------|------------------|
| $ReCoS_{v0}$ | 500 | 2500 | 12 | easy |
| $ReCoS_{v1}$ | 1000 | 5000 | 12 | medium |
| $ReCoS_{v2}$ | 15982 | 5000 | 12 | hard |

In summary, as illustrated in Table 3, we have generated three versions of benchmarks: $ReCoS_{v0}$ comprising 500 original image-text pairs, $ReCoS_{v1}$ extending from $ReCoS_{v0}$ with an additional 500 similar image-text pairs to augment task difficulty, and $ReCoS_{v2}$ extending from $ReCoS_{v1}$ with an additional 14982 pure images resembling those in $ReCoS_{v0}$ to further heighten task difficulty.

## 4 EXPERIMENTS

In this section, we conduct an evaluation of several classic image-text retrieval models on our novel benchmarks. Furthermore, we present a comprehensive analysis of the performance of these models, delving into the intricacies of the various sub-tasks and versions incorporated within our benchmarks.

## 4.1 Experiment Setup

**Benchmarks:** In this experiment, we employ the following five benchmarks:

- **Flickr30K and MS-COCO** benchmarks are derived from everyday life scenarios. The primary distinction lies in the number of test images: the former includes 1,000 images, while the latter encompasses 5,000.
- **Flickr30K-FG and MS-COCO-FG** benchmarks [3] offer more detailed textual descriptions while maintaining the image quantity of their original datasets, $Flickr30k$ and $MS-COCO$.
- **ReCoS** benchmark, introduced in this paper, comprises $1,000$ test images depicting complex real-life scenarios. To assess its performance against previous benchmarks, we conducted comparative experiments using the standardized image-text pairs of $ReCoS_{v1}$. However, in practice, $ReCoS_{v2}$ (with some images lacking annotations) poses significantly greater difficulty than $ReCoS_{v1}$, enabling a more comprehensive evaluation of models' capabilities in image-text retrieval, particularly in fine-grained comprehension of text for image retrieval.

**Baselines and Implementation details:** All experiments are conducted on NVIDIA V100 GPUs, and we use as large models as possible. We used the default configuration in LAVIS [16] for CLIP, ALBEF, BLIP, and BLIP2, and the experimental setup in [39] for X-VLM and X2-VLM. Additionally, VSE++ adhered to the configuration from [9], and SCAN followed that from [14], VSRN followed that from [20].

**Evaluation Metrics:** Following established research practices [9, 15, 20], we evaluate image-text retrieval performance using Recall@k (R@k). This metric quantifies the percentage of queries

accurately retrieving the ground truth within the top-k results, where k is selected from values of 1, 5, and 10.

## 4.2 Results and Analysis

Table 2 displays the experimental results of nine models across the five benchmarks. Initially, we note that the recall rates (R@K) of non-pre-trained models, namely VSE++, SCAN, and VSRN, are notably lower than those of pre-trained models, including CLIP, AL-BEF, X-VLM, X2-VLM, BLIP, and BLIP2. However, we also observed performance declines in both pre-trained and non-pre-trained models specifically on our benchmark $ReCoS_{v1}$ when compared to the other four existing benchmarks. This decrease highlights the challenges in our benchmarks regarding accurately retrieving correct images for a given text description in the presence of numerous similar images, and vice versa. X2-VLM and BLIP2 demonstrated superior performance across most tasks, indicating their effectiveness. Nevertheless, our benchmark poses a substantial challenge even for both X2-VLM and BLIP2, motivating the research community to enhance them in future work.

## 4.3 Sub-tasks Analysis

To investigate the difficulty of different substaks in our benchmark $ReCoC_{v1}$, we choose the representative coarse-grained model method VSE++, fine-grained method SCAN and VSRN, and two pre-trained models CLIP and BLIP2 to perform experiments across the 12 sub-tasks. The results are illustrated in Figure 6 and 7.

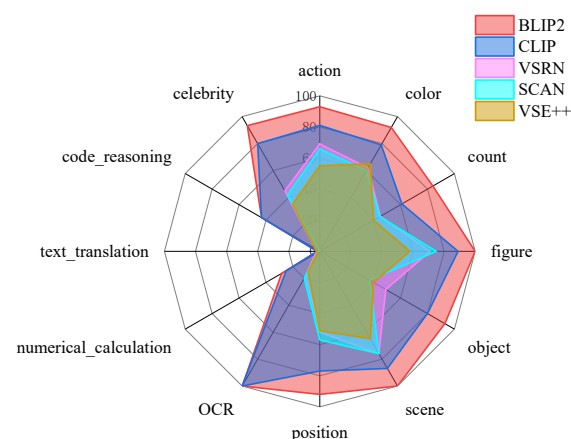

Figure 6: The top-1 retrieval results (R@1) across 12 subtasks in image-to-text (I2T) retrieval task using five models on the ReCoS$_{v1}$ dataset.

In image-to-text and text-to-image retrieval tasks, current models, particularly emphasizing pre-trained models like BLIP2 and CLIP, have showcased exceptional performance. They achieved perfection in figure and scene recognition-based retrieval tasks and consistently maintained over 90% accuracy across five coarse-grained and two fine-grained recognition-based retrieval tasks. The notable superiority of BLIP2 and CLIP, compared to non-pre-trained models, is credited to their robust feature extraction capabilities

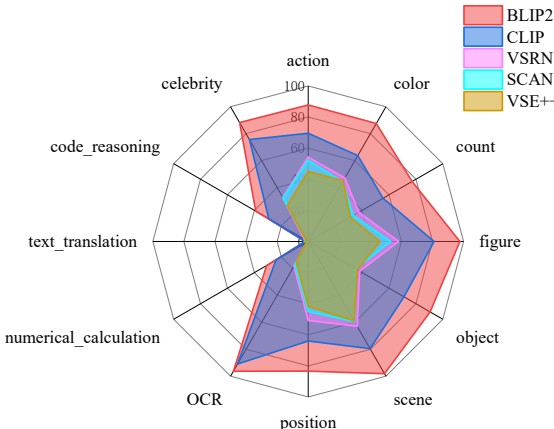

Figure 7: The performance R@1 of 5 models in text-to-image (T2I) retrieval across 12 subcategories on ReCoS$_{v1}$.

from both images and text. Specifically, the BLIP2 model demonstrated nearly flawless performance across various subcategories within the perceptual domain, highlighting its exceptional framework design and extensive training data.

Despite the exceptional performance of models such as BLIP2 and CLIP in perceptual tasks, where they achieve near-perfect results in recognition-based retrieval tasks, particularly for figures and scenes, they face substantial challenges in cognition-based retrieval tasks. A selection of samples from cognition-based retrieval tasks, where both BLIP2 and CLIP struggle, is illustrated in Figure 7. Even in tasks that demand linguistic understanding, such as text-translation-based retrieval, all models fall short, with some achieving zero accuracy. This highlights a limitation in their capacity to effectively integrate perceptual information with deeper cognitive knowledge.

The collective struggle of all models in cognition-based tasks indicates a critical gap in existing methodologies. Our dataset emerges as a valuable resource, prompting a shift in research focus and challenging the community to explore innovative approaches that seamlessly integrate both perceptual and cognitive aspects.

## 4.4 Benchmark Versions Comparative Analysis

In our evaluation, we employed CLIP to assess its retrieval performance across the three versions of our benchmarks, as detailed in Table 3. It is important to note that we refrained from conducting the image-to-text task on $ReCoS_{v2}$ due to the absence of text descriptions for the additional images introduced based on $ReCoS_{v1}$.

The results displayed in Table 3 unveil a consistent decline in performance from $ReCoS_{v0}$ to $ReCoS_{v2}$, signifying a notable escalation in task difficulty. Particularly noteworthy is the more pronounced decrease in R@1 compared to R@10, indicating a heightened challenge in achieving accurate retrieval results. This decline in recall rates sheds light on CLIP's limitations in effectively differentiating between similar images or semantic sentences. It prompts a deeper investigation into enhancing CLIP's discriminative capabilities, specifically tailored to the challenges posed by our benchmark.

**Table 2: Evaluation results of 9 models on 5 benchmarks. Top results for "Image→ Text" and "Text→ Image" tasks are bolded for easy reference. MSCOCO comprises 5k images, whereas both ReCoS and Flickr30k contain only 1k images each.**

| Retrieval task | | | Model | | | | | | | | |
|---|---|---|---|---|---|---|---|---|---|---|---|
| | | | VSE++ | SCAN | VSRN | CLIP | ALBEF | X-VLM | X2-VLM | BLIP | BLIP2 |
| Image→Text | MSCOCO | R@1 | 41.1 | 42.5 | 49.0 | 57.2 | 77.6 | 80.4 | 83.5 | 82.0 | **85.4** |
| | | R@5 | 71.3 | 74.4 | 77.5 | 80.5 | 94.1 | 95.5 | 96.3 | 95.8 | **97** |
| | | R@10 | 81.3 | 85.6 | 87.3 | 87.8 | 94.1 | 98.2 | **98.5** | 98.1 | **98.5** |
| | Filckr30K | R@1 | 52.9 | 65.6 | 66.9 | 86.5 | 77.6 | 96.8 | **98.5** | 96.9 | 97.6 |
| | | R@5 | 80.4 | 88.3 | 89.8 | 98.0 | 94.1 | 99.8 | **100.0** | 99.9 | **100.0** |
| | | R@10 | 87.8 | 93.4 | 94.9 | 99.1 | 97.2 | **100.0** | **100.0** | **100.0** | **100.0** |
| | MSCOCO$_{FG}$ | R@1 | 45.3 | 48.0 | 54.5 | 60.6 | 80.3 | 85.0 | 87.1 | 86.8 | **87.6** |
| | | R@5 | 74.9 | 79.1 | 81.0 | 82.9 | 95.5 | 97.0 | 97.4 | 97.1 | **97.7** |
| | | R@10 | 85.0 | 88.3 | 90.3 | 90.0 | 95.8 | 98.7 | 99.0 | 98.8 | **99.1** |
| | Filckr30K$_{FG}$ | R@1 | 56.2 | 68.9 | 72.6 | 88.7 | 97.2 | 97.4 | **99.1** | 97.1 | 98.5 |
| | | R@5 | 84.5 | 91.0 | 92.4 | 98.3 | 99.8 | 99.8 | **100.0** | **100.0** | **100.0** |
| | | R@10 | 90.3 | 96.0 | 95.9 | 99.0 | **100.0** | **100.0** | **100.0** | **100.0** | **100.0** |
| | ReCoS$_{v1}$ | R@1 | 26.6 | 31.6 | 35.0 | 67.5 | 74.2 | 74.3 | 79.3 | 80.0 | **80.8** |
| | | R@5 | 50.5 | 55.1 | 57.2 | 88.6 | 84.0 | 82.8 | 89.8 | **90.3** | 89.9 |
| | | R@10 | 60.8 | 62.8 | 64.2 | **92.5** | 85.9 | 84.0 | 89.0 | **92.5** | 92.0 |
| Text→Image | MSCOCO | R@1 | 30.3 | 33.0 | 35.2 | 36.5 | 61.0 | 63.1 | 66.2 | 64.5 | **68.3** |
| | | R@5 | 59.4 | 63.1 | 65.2 | 60.8 | 84.5 | 85.7 | 87.1 | 86.0 | **87.7** |
| | | R@10 | 72.5 | 75.1 | 76.3 | 71.0 | 90.7 | 91.6 | 92.2 | 91.7 | **92.6** |
| | Filckr30K | R@1 | 39.6 | 41.2 | 49.4 | 67.0 | 61.0 | 86.1 | **90.4** | 87.5 | 89.7 |
| | | R@5 | 69.9 | 71.5 | 77.1 | 88.9 | 84.5 | 97.4 | **98.2** | 97.6 | 98.1 |
| | | R@10 | 79.6 | 80.7 | 84.4 | 93.3 | 90.7 | 98.7 | **99.3** | 98.9 | 98.9 |
| | MSCOCO$_{FG}$ | R@1 | 34.7 | 37.2 | 39.9 | 39.2 | 64.2 | 68.9 | 70.1 | 68.1 | **72.6** |
| | | R@5 | 64.2 | 66.6 | 69.0 | 64.0 | 86.9 | 88.0 | 89.6 | 88.5 | **90.2** |
| | | R@10 | 77.1 | 77.9 | 80.1 | 73.7 | 91.3 | 93.0 | **95.9** | 93.7 | 94.2 |
| | Filckr30K$_{FG}$ | R@1 | 44.7 | 45.6 | 54.1 | 71.3 | 90.1 | 90.0 | **92.8** | 91.6 | 92.5 |
| | | R@5 | 74.8 | 76.3 | 80.9 | 90.6 | 98.6 | 98.5 | **99.5** | 98.6 | 99.0 |
| | | R@10 | 83.5 | 84.7 | 87.6 | 94.7 | 99.5 | 99.2 | **99.9** | 99.4 | 99.4 |
| | ReCoS$_{v1}$ | R@1 | 19.6 | 20.5 | 27.0 | 54.2 | 67.6 | 68.3 | **79.3** | 73.7 | 74.9 |
| | | R@5 | 42.1 | 43.7 | 50.9 | 81.5 | 81.6 | 80.1 | **89.1** | 88.7 | 88.4 |
| | | R@10 | 52.5 | 52.8 | 59.6 | 87.8 | 84.1 | 81.5 | 89.5 | **90.9** | 90.7 |

**Table 3: Comparison results for CLIP across three versions of our benchmark.**

| Test Mode | Image → Text | | | Text → Image | | |
|---|---|---|---|---|---|---|
| | R@1 | R@5 | R@10 | R@1 | R@5 | R@10 |
| ReCoS$_{v0}$ | 83.0 | 92.4 | 94.6 | 70.6 | 88.2 | 92.9 |
| ReCoS$_{v1}$ | 67.5 | 88.6 | 92.5 | 54.2 | 81.5 | 87.8 |
| ReCoS$_{v2}$ | - | - | - | 25.0 | 50.1 | 60.1 |

## 5 CONCLUSIONS

In this study, we scrutinize common benchmarks for image-text retrieval and find that they fail to fully assess the true capabilities of fine-grained cross-modal semantic alignment due to the coarse granularity of images and texts, excessive descriptive information, and overly simplistic retrieval scenarios. Therefore, We propose *ReCoS*, a novel image-text benchmark, designed to address challenges in cross-modal retrieval in real-life scenarios, with three versions: $ReCoS_{v0}$, $ReCoS_{v1}$, and $ReCoS_{v2}$.To compare with existing benchmarks, we select the moderately challenging $ReCoS_{v1}$ as the standard benchmark, where each image is annotated. By evaluating representative image-text retrieval models on *ReCoS*, we demonstrate and analyze the models' fine-grained semantic understanding capabilities across various subtasks through extensive experiments. The experimental results indicate that even state-of-the-art retrieval models exhibit certain limitations in real-world scenarios. In the future, we aim to extend more complex retrieval subtasks, such as complex charts, intricate flowcharts, with the hope that the novel benchmark will inspire further research into cross-modal retrieval.

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
