# OpenReview forum: "ReCoS: A Novel Benchmark for Cross-Modal Image-Text Retrieval in Complex Real-Life Scenarios"
_acmmm.org/ACMMM/2024/Conference — MM2024 Poster_

### Official Review · Reviewer_1Nvs · 2024-05-20

**Rating:** 2
**Confidence:** 3

**Summary:**

This paper proposes a novel benchmark for the task of cross-modal retrieval, it addresses 12 retrieval subtasks and contains three variants.

**Strengths:**

The author conducted thorough evaluations of existing methods on said benchmark and compared the results with those of existing datasets. The paper is well-written, in that it conveys the motivation and methodology of the proposed benchmark clearly.

The benchmark carries several merits compared to the existing ones, such as the inclusion of assessing the models' confidence through entropy.

**Limitations:**

**Novelty and the amount of contribution**

While the benchmark is insightful and aligns well with the motivation, I am afraid that it alone may not constitute a sufficiently novel submission.

Indeed, this entire paper surrounds the benchmark, with no additional contributions on the task in terms of, e.g., method or other detailed analysis on the performance (4.3 includes some discussions on the sub-tasks and their performance, though I personally find it not enough to be claimed as a major contribution).


**The diversity of the sub-tasks**

As the authors show in Figure 7, with the ReCos$_{v1}$ metric the current models suffer greatly from the three sub-tasks, *code reasoning*, *text translation* and *numerical calculation*, meanwhile performing relatively well (take BLIP2 for instance) on the remaining ones.

To an extent, this result is to be expected because cognition-based sub-tasks are inherently harder, and one might even argue that they are entirely on a different landscape -- which brings me to my concern as to why they shall be mixed with all the other recognition-based sub-tasks. If one wishes for a retrieval model to excel at text translation, for example, one should either aim to build an AGI (alas we are not there yet), or enlist a translation model to help instead of relying on the retrieval mode alone.

All in all, I am not against evaluating a retrieval model on different sub-tasks, I am just not convinced that the author's choice of the subtasks is entirely justifiable -- and to consider that this benchmark, once released, shall encourage others to test and develop on it, one need to consider the implications and impact this benchmark will bring to future research.

**The current state-of-the-art on the proposed benchmark**

Given that part of the author's motivation is to address the fact the state-of-the-art methods on the existing metrics have reached near-perfection, the newly proposed benchmark needs to be more challenging. Indeed, in Table 2 we observe a noticeable difference in the values. However, this is compared with the *v1* metric, I am a bit curious about how the *v0* compares.

Also, a note that even with the *v1* metric, the performance difference compared to the existing benchmarks is not extremely large, which brings two questions:

 * Will this newly proposed benchmark last long before it reaches the point where it can no longer distinguish state-of-the-art models (just like the current ones)?
* How much of the performance drop is contributed by the much harder cognitive-based sub-tasks discussed above?

**Minor comments**

It would be great if the author could provide some information on the inference cost of the three new metrics, especially the time that takes for one evaluation.

**Suitability:**

3

---

### Official Review · Reviewer_c8aP · 2024-05-23

**Rating:** 5
**Confidence:** 3

**Summary:**

The paper introduces ReCoS, a novel benchmark for cross-modal image-text retrieval designed to address the limitations of existing benchmarks. ReCoS is tailored for complex real-life scenarios and emphasizes cognition-based tasks, aiming to provide a more holistic assessment of model capabilities. The benchmark includes 12 retrieval tasks, focusing on three cognition-based tasks, and utilizes original data sources to ensure novelty and minimize data leakage risks. The paper also presents three versions of the benchmark with varying difficulty levels and evaluates several representative image-text retrieval models on these benchmarks. The results highlight the challenges faced by existing methods, particularly in cognition-based retrieval tasks, and underscore the need for innovative approaches to handle real-world complexities. However, the authors should consider providing more insights into how the benchmark can be generalized and ensuring that it does not inadvertently favor certain model architectures or tasks. Additionally, a discussion on the potential computational requirements and resource availability for using ReCoS would be beneficial for the research community.

**Strengths:**

1.Comprehensive Benchmark Design: ReCoS covers a wide range of subtasks, going beyond object recognition to include aspects like color, count, position, and complex retrieval tasks such as OCR, code image retrieval, and text translation.
2.Original Data Sources: The use of original data sources helps to ensure the novelty of the benchmark and reduces the risk of overfitting to existing datasets.
3.Balanced Text Descriptions: The benchmark strikes a balance between overly detailed and under-detailed text descriptions, which is crucial for creating a fair and challenging evaluation environment.
4.Real-World Complexity Alignment: ReCoS is designed to reflect the intricacies of real-world scenarios, providing a more accurate assessment of Visual-Language Pretraining models.
5.Incremental Difficulty Levels: The creation of three versions of the benchmark with varying difficulty levels allows for a tiered evaluation of model capabilities.
6.Thorough Evaluation: The paper includes an extensive evaluation of several state-of-the-art models, providing insightful analysis into their performance across different tasks.

**Limitations:**

1.Complexity of Benchmark: While the complexity of ReCoS is a strength, it may also be a weakness as it could potentially make the benchmark too challenging for current models, possibly slowing down research progress in the short term.
2.Resource Intensity: The use of original data sources and the creation of a comprehensive benchmark may require significant resources, which could limit accessibility for smaller research groups.
3.Potential for Bias: As with any new benchmark, there is a risk of introducing bias towards certain types of models or tasks. Ensuring a wide representation of model architectures and tasks is crucial.
4.Generalization Concerns: The paper does not extensively discuss how well the results from ReCoS generalize to other real-world scenarios outside the benchmark.
5.Overemphasis on Cognition-Based Tasks: While cognition-based tasks are important, there might be an argument for a more balanced approach that equally emphasizes perceptual tasks, given their relevance in many real-world applications.
6.Evaluation Methodology: The paper could benefit from a more detailed discussion on the evaluation methodology, including how the models were selected and how their performance was compared.

**Suitability:**

3

---

### Official Review · Reviewer_WVYz · 2024-05-25

**Rating:** 4
**Confidence:** 3

**Summary:**

In this paper, the authors propose a more robust evaluation benchmark specifically designed for cross-modal image-text retrieval in complex real-life scenarios. The delineate four essential characteristics that a comprehensive evaluation benchmark for image-text retrieval tasks should possess. Furthermore, the authors evaluated several representative image-text retrieval models on their proposed benchmark dataset, including both non-pre-trained models and pre-trained models, to further analyze their capabilities in complex real-world scenarios.

**Strengths:**

1、The article summarizes four essential characteristics that a foundational dataset for image-text retrieval tasks should possess and introduces a new benchmark dataset. The introduction of this dataset promotes further development in the field of multimodal retrieval.
2、The organization of the article is excellent, and the construction process of the dataset is described in detail.

**Limitations:**

1、The dataset proposed by the authors has three different versions, but during the experiments, only the results on versions ReCoSv1 and ReCoSv2 were observed. Therefore, is the smaller dataset version ReCoSv0 useful?
2、The new dataset proposed by the authors is relatively small, and the specific division of the training set, validation set, and test set is unclear.
3、Regarding the experimental results in Table 2, there is a question about how these results were obtained. On which datasets were the various comparison models trained, especially the experimental results for ReCoSv1?
4、The authors' experimental analysis is not particularly clear. I believe that the potential improvement points of existing image-text retrieval methods should be analyzed in detail based on the experimental results and the proposed new dataset.
5、The authors should provide more detailed samples from the newly created dataset. The samples shown in Figure 2 are too simplistic.

**Suitability:**

3

---

### Official Review · Reviewer_RATs · 2024-05-26

**Rating:** 1
**Confidence:** 3

**Summary:**

This paper focuses on the task of image-text retrieval. It is argued that with the rapid development of visual language pre-training models, the existing image-text datasets are relatively homogeneous in composition and cannot objectively and correctly evaluate the performance of the models from multiple perspectives. Therefore, this paper proposes an evaluation criterion called ReCoS in this area, which collects and generates new data to challenge existing graphic-text retrieval methods for more fine-grained and complex scenarios.

**Strengths:**

a.	Compared with previous methods for improving benchmarks, this paper introduces a variety of additional tasks such as code image retrieval, landmark construction retrieval and celebrity retrieval to make the evaluation results more comprehensive.
b.	This paper proposes an easier-to-use quantitative method for assessing the quality of generated text descriptions.The method proposed in this paper is easy to follow.
c.	This paper presents several different versions of benchmarks that can be effectively evaluated on a scale of difficulty from easy to hard.

**Limitations:**

a.	In the validation of of text description module ,the description of how to use the cross-entropy function to calculate dis(t) is not clear, and there is no way to know whether additional terms are added to expand the range of dis(t) to [0,logk], and the introduction of hyperparameters is not analyzed with appropriate sensitivity,and it is difficult to obtain useful information in the upper part of Figure 5.
b.	In the image confusion module mentioned in this article, for example, “(2) Generate descriptions ... (3) Attempt to ...” It is not possible to obtain information on the specific methodology and criteria used for the assessment, and it would be appreciated if the relevant material could be supplemented.
c.	Among the benchmarks proposed in this paper, the amount of data in “ReCoSv1”, which can be used for the evaluation of bi-directional retrieval is on the low side, which weakens the persuasive power of the results to a certain extent.

Typos and minors
The whole manuscript should be carefully checked for typos, grammar and syntax, as there are some of them. The following are some example:
a.	In lines 581 and 582, that is, “Specifically, we identify descriptions... .logk].text descriptions...” section. The period before “text” here should be replaced with a comma, otherwise the break could be easily misinterpreted.
b.	Suggest revising the logic of the article, the  readablity of this paper is weak.

**Suitability:**

3

---

### Meta-Review · Area_Chair_p4Zv · 2024-07-02

**Recommendation:** Accept (Poster)
**Confidence:** 1

**Metareview:**

Except for one reviewer, there is a general consensus to accept this paper.
The authors' rebuttal successfully addressed most of the concerns raised by the reviewers.
Some typos and minors were noted. Please, in case of acceptance double check.
The major strength mentioned by the reviewers is the usefulness of the proposed novel benchmark dataset for image-text retrieval. The introduction of this dataset promotes further development in the field of multimodal retrieval.
The organization of the article is excellent, and the construction process of the dataset is described in detail.